# Synthesis of Hierarchical Porous Carbon in Molten Salt and Its Application for Dye Adsorption

**DOI:** 10.3390/nano9081098

**Published:** 2019-07-31

**Authors:** Saisai Li, Haijun Zhang, Shiya Hu, Jie Liu, Qing Zhu, Shaowei Zhang

**Affiliations:** 1The State Key Laboratory of Refractories and Metallurgy, Wuhan University of Science and Technology, Wuhan 430081, China; 2College of Engineering, Mathematics and Physical Sciences, University of Exeter, Exeter EX4 4QF, UK

**Keywords:** hierarchically porous carbon, molten salt method, Fe(NO_3_)_3_·9H_2_O, dye adsorption

## Abstract

Hierarchical porous carbon was successfully synthesized from glucose in a molten salt at 800 °C for 2 h. It was amorphous and contained numerous oxygen containing functional groups on its surface. The porous carbon with 1.0 wt% Fe(NO_3_)_3_·9H_2_O oxidizing agent showed the highest specific surface area of 1078 m^2^/g, and the largest pore volume of 0.636 cm^3^/g, among all of the samples. Raman and TEM results revealed that it had more defects and pores than other as-prepared carbon materials. The adsorption capacities of as-prepared porous carbon for methylene blue (MB) and methyl orange (MO) were 506.8 mg/g and 683.8 mg/g, respectively. The adsorption isotherms fit the Langmuir model and the adsorption kinetics followed the pseudo-second-order kinetic model.

## 1. Introduction

Dyes have long been used in plastic, paper, leather, textile, and other industrial sectors. Wastewaters discharged from these industrial sectors often contain a certain amount of dyes, especially water-soluble organic dyes, due to their low biodegradability [1,2,3]. The conventional biological treatment processes fail to effectively treat these dye wastewaters, due to which, several physical or chemical processes, including osmosis [4], electro flotation [5], chemical oxidation [6], ion exchange [7], and filtration [8], have been developed in recent years. Unfortunately, these processes are not economically-viable and are still not sufficiently effective for treating a wide range of dye wastewaters [9]. Physical adsorption has become the most popular method so far for removing dyes from wastewaters, owing to its low cost, high efficiency, easy operation, diversity in adsorbents, and high stability towards the adsorbents.

Porous carbon materials have been widely applied for removing dyes and other organic and inorganic pollutants from drinking water due to their high specific surface area [10,11,12]. In particular, those with hierarchical porosities (micro-/meso-/macropore) exhibit enhanced dye adsorption performance. This is because that the hierarchical pore structures can provide various functions: Micropores provide high surface area and large adsorption capacity, mesopores facilitate dye transporting, and macropores act as reservoirs for dye molecules.

Several techniques, mainly physical or chemical activation combined with templating, have been used to synthesize hierarchical porous carbon materials [13,14,15]. However, they exhibited some disadvantages, such as high carbonization temperature, multiple steps, low carbon yield, vessel corrosion, and environmental pollution. Recently, molten salt synthesis methods have become applicable to the preparation of a wide range of nanomaterials, including binary, ternary/multinary oxides [16,17,18,19,20,21,22,23,24,25], hydroxide [26], non-oxides [27], and porous carbon materials [28,29,30,31]. For example, Liu et al. synthesized porous carbon and carbon sheets in molten LiCl/KCl containing different oxysalts (such as Fe(NO_3_)_3_·9H_2_O, KClO_3_, and K_2_CO_3_), which facilitated pore formation in the final porous carbon [32]. Deng et al. prepared nitrogen-doped hierarchically porous carbon materials with high specific surface area in molten ZnCl_2_ [33]. Despite these studies, there have been few studies on the preparation of porous carbon with high specific surface area by using a combined activator, ZnCl_2_ and Fe(NO_3_)_3_·9H_2_O.

In this work, hierarchical porous carbon was synthesized in ZnCl_2_-KCl molten salt using a low-cost and eco-friendly glucose carbon source and an Fe(NO_3_)_3_·9H_2_O oxidizing agent. The effects of additional amounts of Fe(NO_3_)_3_·9H_2_O on the specific surface area of as-prepared samples were investigated, as were their adsorption capacities for methylene blue (MB) and methyl orange (MO). 

## 2. Materials and Methods

### 2.1. Raw Materials and Sample Preparation

The main starting materials included: Glucose (C_6_H_12_O_6_·H_2_O, AR; Bodi chem. Co. Ltd. Tianjin, China), zinc chloride (ZnCl_2_, AR; Sinopharm chem. Co. Ltd. Shanghai, China), potassium chloride (KCl, AR; Sinopharm chem. Co. Ltd. Shanghai, China), commercial ferric nitrate (Fe(NO_3_)_3_·9H_2_O, 99.0%, Lia Chemical Co., Ltd. Wuhan, China), commercial ferric chloride (FeCl_3_·6H_2_O, 99.0%, Lia Chemical Co., Ltd. Wuhan, China), zinc nitrate (Zn(NO_3_)_2_, AR; Sinopharm chem. Co. Ltd. Shanghai, China), methylene blue trihydrate (C_16_H_18_ClN_3_S·3H_2_O, Sinopharm chem. Co., Lth. Shanghai, China), and methyl orange (C_14_H_14_N_3_SO_3_Na, Sinopharm chem. Co., Lth. Shanghai, China). 

In a typical preparation process, 2 g glucose and different amounts of Fe(NO_3_)_3_·9H_2_O (0–2.0 wt% Fe of carbon amount, the samples are referred to as 0 Fe, 0.5 Fe, 1.0 Fe, 1.5 Fe, and 2.0 Fe, respectively) were milled together and further combined with 20 g of KCl and ZnCl_2_ (in the weight ratio of 1:2. The eutectic point of the binary salt is 234 °C.). The mixed powder was contained in an alumina crucible and heated at 2 °C/min to 800 °C and soaked for 2 h in an alumina tube furnace protected by flowing Ar (99.999 vol% pure). The reacted sample was subjected to repeat water-washing before being oven-dried for 12 h at 80 °C.

### 2.2. Adsorption Test

Adsorption processes were investigated using MB and MO. Typically, 20 mg of as-prepared sample (porous carbon) powder was added into 50 mL MB aqueous solution with a concentration ranging from 0 to 500 mg/L. The suspension was then homogenized for 30 min at 25 °C using a magnetic stirrer. After centrifugation, the solution part was examined using a UV-visible spectrophotometer.

To study the adsorption kinetics, 60 mg of as-prepared porous carbon were added to 150 mL aqueous dye solution (100 mg/L) in a beaker. The suspension was stirred using a magnetic stirrer. After a certain time period, 5 ml of the solution was taken and centrifuged for UV-vis examination. The maximum adsorption wavelengths of 665 nm and 464 nm, in the cases of MB and MO, were used respectively for calculation.

The equilibrium adsorption amount of the porous carbon materials was determined based on the following equation:(1)$$q_{e}=\frac{(c_{0}-c_{e})v}{m}$$
where *q_e_* (mg/g) is the equilibrium adsorption amount, *c_0_* (mg/L) is the initial concentration of dye solution, *c_e_* (mg/L) is the equilibrium concentration of dye solution, *v* (L) is the volume of dye solution, and *m* (g) is the mass of porous carbon.

### 2.3. Characterization

Powder X-ray diffraction (XRD) analysis was carried out using a Philips X’Pert PRO diffractometer (Xpertpro, PHILIPS, Hillsboro, The Netherlands) at 40 mA and 40 kV and with a Cu K*α* radiation (*λ* = 0.1542 nm). The scan range was between 10° and 90° (2θ) at 40 mA and 40 kV and the scan rate was 2°/min with a step size of 0.05°. Microstructure and phase morphologies of as-prepared samples were examined by using a field-emission scanning electron microscope (FE-SEM; Nova400NanoSEM, 15 kV, Philips, Amsterdam, The Netherlands) and a transmission electron microscope (TEM) JEM-2100UHRSTEM, 200 kV (JEOL, Tokyo, Japan) along with an energy dispersive spectrometer (EDS) Penta FET X-3 Si (Li) (EDS, IET 200, Oxford, UK). The specific surface area of as-prepared samples was calculated based on Brunauere Emmette Teller (BET) nitrogen adsorption/desorption analysis carried out using a gas sorption analyzer Autosorb-1-MP/LP (Quantachrome, Boynton Beach, FL, USA). Functional groups on the surface of a sample were identified by Fourier Transform Infrared Spectroscopy (FTIR) VERTEX 70 (Bruker, Karlsruhe, Germany). The adsorption of MB or MO was examined by Ultraviolet–visible spectroscopy UV-2550 (Shimadzu Corporation, Kyoto, Japan). Raman spectra were recorded using a Horiba Jobin-Yvon Labram-HR800 Raman spectrometer (Raman, Paris, France) with an excitation wavelength of 532 nm. 

## 3. Results and Discussion

### 3.1. Microstructural Characterization of As-Prepared Porous Carbon

Powder X-ray diffraction patterns of samples resultant from 2 h firing in molten salt at 800 °C are shown in Figure 1a, revealing a broad diffraction peak in each case, centered at approximately 26° and corresponding to the (002) lattice plane of graphite, and indicating the amorphous nature of the carbon formed in the samples. Apart from the amorphous carbon, Fe_2_O_3_ and Fe_3_O_4_ were identified in the samples using Fe(NO_3_)_3_·9H_2_O, suggesting the decomposition of Fe(NO_3_)_3_·9H_2_O into Fe_2_O_3_ firstly and then some Fe_2_O_3_ became Fe_3_O_4_ by reductive carbon coming from glucose upon firing [34,35]. The obtained lattice parameter values for Fe_2_O_3_ were, respectively, *a* = 13.21 Å, *b* = 9.78 Å, and *c* = 8.37 Å, and *a* = 6.21 Å, *b* = 5.89 Å and *c* = 14.77 Å for Fe_3_O_4_. Fourier transform infrared spectroscopy were used to identify functional groups in as-prepared carbons with various amounts of Fe(NO_3_)_3_·9H_2_O (Figure 1b). The main bands at 1316 and 1617 cm^-1^ were attributed to the C=C stretching vibration, and the other two bands at 3470 and 1380 cm^-1^ corresponded to the O–H (hydroxyl or carboxyl) stretching vibration and the C–OH stretching vibration, respectively, and this later verified the existence of aldehyde. The above observations indicated the existence of oxygen containing functional groups on the carbon formed in as-prepared samples, which favored the adsorption of the dyes [36].

Scanning electron microscope images of as-prepared carbon samples resulting from 2 h firing at 800 °C in the molten salt with various amounts of Fe(NO_3_)_3_·9H_2_O are shown in Figure 2, and reveal the formation of irregular carbon particles and carbon sheets. With increasing Fe content, the morphology of the carbon particles changed very little. The carbon particles resulted from the decomposition of glucose and the nucleation of intermediates and their subsequent growth during carbonization, whereas the carbon sheets were formed via the merging and growing of carbon particles facilitated by highly reactive lithium, potassium, and chloride ions in the molten salt medium [37,38]. 

N_2_ adsorption–desorption isotherms of as-prepared carbon samples, along with the derived pore size distributions are presented in Figure 3. Similar type I isotherms were seen in all cases, and the adsorption amounts within different relative pressure ranges reflected different pore structures (micropores (P/P_0_ between 0−0.1), mesopores (P/P_0_ between 0.1−0.8), and macroporous (P/P_0_ between 0.8–1.0) [39,40]. Therefore, the N_2_ sorption profiles evidently reveal that the as-prepared carbon materials possessed a hierarchical micro-/mesoporous structure. From Figure 3a, it can be seen that the sample with 1.0 wt% Fe showed the maximum adsorption volume, indicating it has the largest specific surface area. Figure 3b shows the pore size distribution curves derived from the desorption isotherms using the Barret–Joner–Halenda (BJH) method, which reveal the formation of mainly micro- and meso-porosities in as-prepared carbon materials. The porosity parameters of as-prepared porous carbon materials are listed in Table 1. As can be seen from Table 1, the BET specific surface areas of all of the samples with Fe(NO_3_)_3_·9H_2_O increased compared to the sample without Fe(NO_3_)_3_·9H_2_O, and upon increasing the amount of Fe(NO_3_)_3_·9H_2_O to 1.0 wt%, the sample showed the highest BET specific surface area of 1078 m^2^/g and the highest pore volume of 0.636 cm^3^/g. Furthermore, the mesoporous volume percent reached up to 60% with increasing the Fe amount to 1.0–1.5 wt%, which was probably mainly due to the decomposition of Fe(NO_3_)_3_·9H_2_O during carbonization [41].

Figure 4 presents Raman spectra of porous carbon materials prepared at 800 °C for 2 h with various amounts of Fe. Two main peaks at 1350 and 1590 cm^-1^ were respectively assigned to the D band and G band [42]. With the increase of the Fe amount, the value of I_D_/I_G_ increased initially and then decreased, demonstrating the existence of many defects in the as-prepared porous carbon.

Based on the above test results, it can be concluded that addition of 1.0 wt% Fe(NO_3_)_3_·9H_2_O, resulted in the largest specific surface area. In order to compare and further reveal the effectiveness of Fe(NO_3_)_3_·9H_2_O, samples using respectively FeCl_3_·6H_2_O and Zn(NO_3_)_2_ as oxidizing agents were also prepared under similar processing conditions to those in the case of using 1.0 wt% Fe(NO_3_)_3_·9H_2_O. Amorphous carbon was also formed in these two cases (Appendix A). For the sample with added FeCl_3_·6H_2_O, one peak at about 36.2° appeared, which was corresponded to Fe_3_O_4_. As shown in Appendix A, porous carbons prepared using FeCl_3_ and Zn(NO_3_)_2_ as oxidizing agents were mainly comprised of carbon particles. Different from the case of using Fe(NO_3_)_3_·9H_2_O, carbon sheets were only occasionally seen. FTIR identified the existence of some oxygen containing functional groups and C–H groups on the surface of as-prepared porous carbon (Appendix A). The specific surface areas of the samples prepared with FeCl_3_ and Zn(NO_3_)_2_ were 534 and 654 m^2^/g, respectively (Appendix A), which are even lower than that of the sample prepared without using any oxidizing agents (Table 1), indicating that neither FeCl_3_ nor Zn(NO_3_)_2_ was useful for increasing the specific surface area. The sample with Zn(NO_3_)_2_ had a higher surface area than in the case of using FeCl_3_, which was attributed to the oxidizability of Zn(NO_3_)_2_. The samples prepared with Fe(NO_3_)_3_·9H_2_O showed the highest surface area, since Fe(NO_3_)_3_·9H_2_O can act as an effective oxidizing agent during carbonization [43]. It reacts with glucose and releases some gaseous phases, such as NO_x_, CO, and CO_2_, facilitating the formation of micro/mesopores [41]. Comparison of the Raman results (Appendix A and Figure 4) also reveals that the samples with Fe(NO_3_)_3_·9H_2_O had a higher I_D_/I_G_ value, suggesting the existence of more defects in them.

In order to further reveal the porous structures of the as-prepared carbon materials, Figure 5, as an example, shows TEM images and EDS results of the sample prepared with 1.0 wt% Fe. As seen from Figure 5a,b, the as-prepared carbon contained a large number of mesopores, which were consistent with the BET results (Figure 3). EDS of the selected region in Figure 5a indicated that the sample was mainly composed of C, Fe, and O elements. 

### 3.2. Adsorption Performance of As-Prepared Porous Carbon for Methylene Blue and Methyl Orange

The adsorption behaviors of porous carbon materials prepared with various amounts of Fe (Figure 1, Figure 2, Figure 3, Figure 4 and Figure 5) were also examined using MB and MO as adsorbates, and the adsorption isotherms were simulated using the Langmuir and Freundlich models expressed as follows:(2)Langmuir isotherm:      qe=Q0bce1+ce1n
(3)Freundlich isotherm:      qe=kFce1n
where *q_e_* (mg/g) is the equilibrium adsorption amount, *Q*_0_ (mg/g) is the maximum adsorption amount, *b* (L/mg) is the constant term related to the energy of adsorption, *c_e_* (mg/L) is the equilibrium concentration of dye solution, and *k_F_* and *n* are the Freundlich constants.

The calculated adsorption parameters and correlation coefficients (*R^2^*) (Table 2 and Table 3) suggest that the Langmuir model fits the data better than the Freundlich model. The maximum monolayer adsorption capacity (*Q*_0_) of the as-prepared porous carbon (1.0 Fe) for MB and MO were calculated to be 506.8 and 683.8 mg/g, respectively, which was mainly attributed to the much larger specific surface area of the hierarchical porous carbon materials and the oxygen containing functional groups existing on their surfaces. Table 4 compares the adsorption capacity for dyes of hierarchical porous carbon materials and other adsorbents. The high adsorption capacity of hierarchical porous carbon materials prepared in this work indicates their feasibility in dye removal. 

The following pseudo-first-order and pseudo-second-order kinetic models [1] were also used to assist understanding the relevant adsorption mechanism:(4)$$\text{Pseudo-first-order}\;\mathrm{model}:\;q_{t}=q_{e}(1-e^{-k_{1}t})$$
(5)$$\text{Pseudo-second-order}\;\mathrm{model}:\;\frac{t}{q_{t}}=\frac{1}{\kappa_{2}q_{e}^{2}}+\frac{t}{q_{e}}$$
where *q_t_* (mg/g) is the adsorption amount at time *t*, *k_1_* and *k_2_* (g·mg^−1^·min^−1^) are respectively the pseudo-first-order and pseudo-second-order rate constants.

As indicated by the results listed in Table 2 and Table 3, the pseudo-second-order model shows a good linearity, with correlation coefficients (*R*^2^) ≥ 0.99, suggesting that the adsorption kinetics of as-prepared porous carbon follow this model (Figure 6).

## 4. Conclusions

A simple molten salt method was used to synthesize hierarchical porous carbon at 800 °C for 2 h. Glucose and ZnCl_2_-KCl were used as the carbon source and reaction medium, respectively. Numerous oxygen containing functional groups existed on the surface of the as-prepared porous carbon. The porous carbon prepared with 1.0 wt% Fe(NO_3_)_3_·9H_2_O addition showed the highest specific surface area of 1078 m^2^/g and the largest pore volume of 0.636 cm^3^/g, which were attributed to Fe(NO_3_)_3_·9H_2_O acting as an oxidizing agent during carbonization, favoring the formation of micro/mesopores. The adsorption capacities of as-prepared hierarchical porous carbon for methylene blue and methyl orange were respectively 506.8 mg/g and 683.8 mg/g. The adsorption isotherms fit the Langmuir model and the adsorption kinetics followed the pseudo-second-order kinetic model.

## Figures and Tables

**Figure 1 nanomaterials-09-01098-f001:**
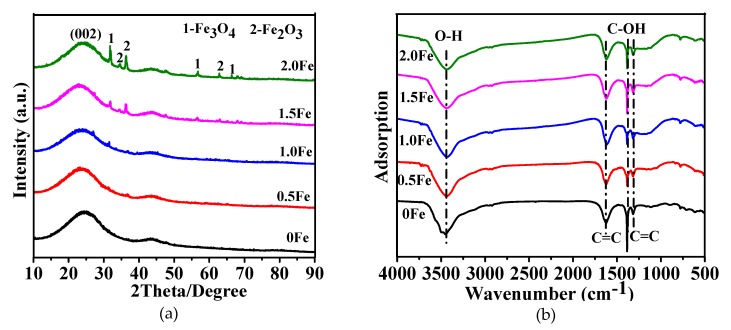
(**a**) Powder X-ray diffraction (XRD) pattern and (**b**) Fourier transform infrared spectroscopy (FTIR) of as-prepared samples with various amounts of Fe(NO_3_)_3_·9H_2_O. (ICDD: 00-016-0653 (Fe_2_O_3_) and 01-076-0956 (Fe_3_O_4_)).

**Figure 2 nanomaterials-09-01098-f002:**
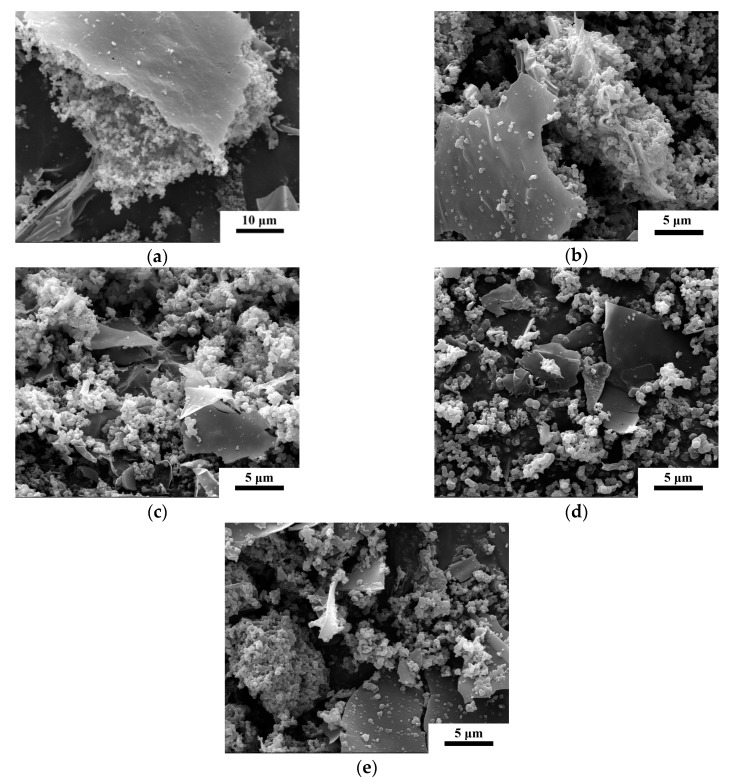
Scanning electron microscope (SEM) images of as-prepared samples whose XRD patterns are shown in Figure 1a. (**a**) 0 Fe, (**b**) 0.5 Fe, (**c**) 1.0 Fe, (**d**) 1.5 Fe, (**e**) 2.0 Fe.

**Figure 3 nanomaterials-09-01098-f003:**
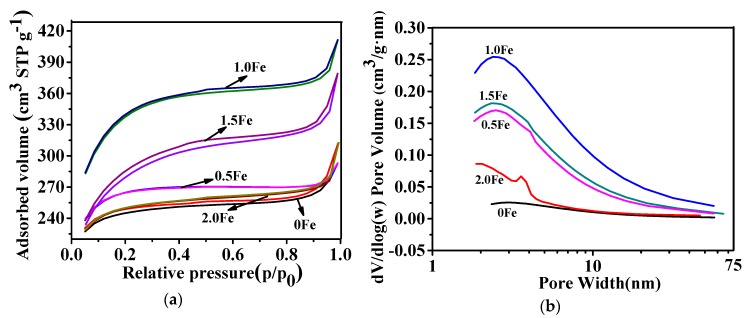
(**a**) N_2_ adsorption–desorption isotherms and (**b**) pore size distribution curves of porous carbon prepared at 800 °C for 2 h with various amounts of Fe.

**Figure 4 nanomaterials-09-01098-f004:**
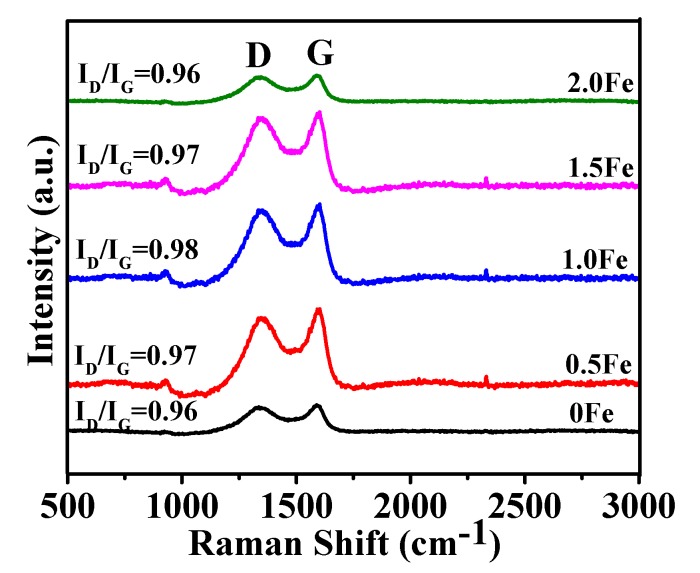
Raman spectra of porous carbon prepared at 800 °C for 2 h with various amounts of Fe.

**Figure 5 nanomaterials-09-01098-f005:**
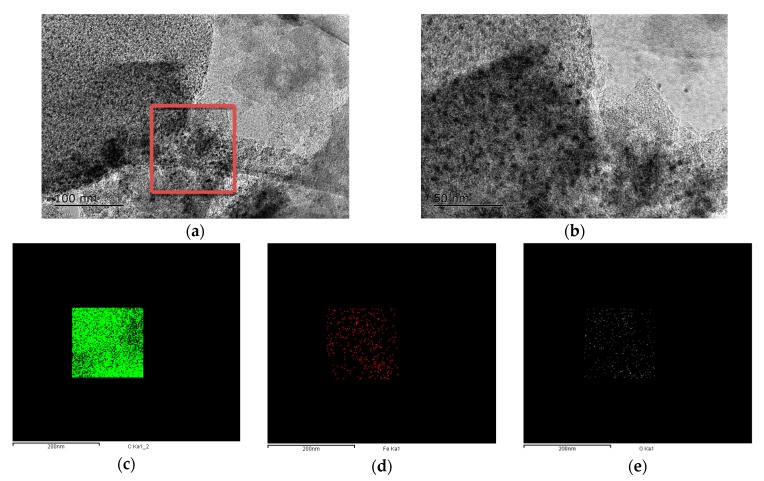
Transmission electron microscope (TEM) and energy dispersive spectrometer (EDS) results of the sample prepared with 1.0 wt% Fe. (**a**) Low-resolution image, (**b**) high-resolution image, (**c**) mapping scan of C in red box, (**d**) mapping scan of Fe in red box, and (**e**) mapping scan of O in red box.

**Figure 6 nanomaterials-09-01098-f006:**
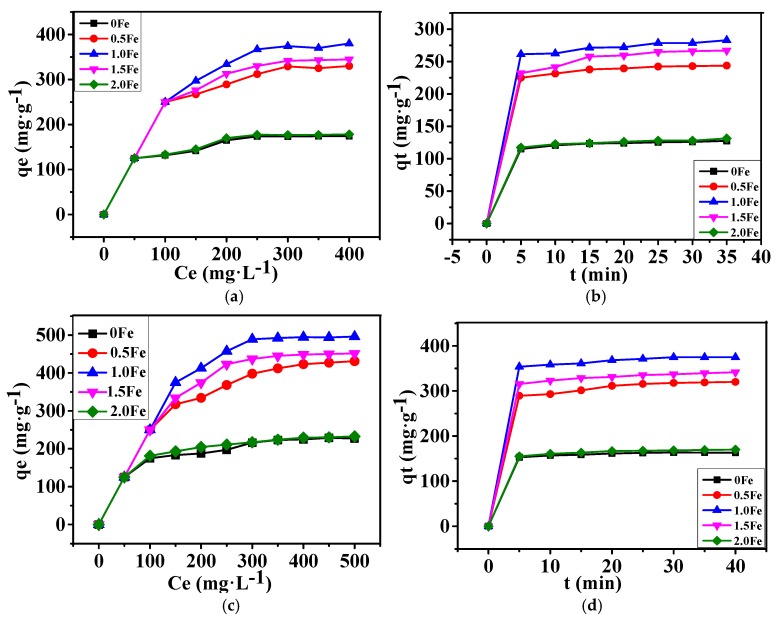
Adsorption isotherms and kinetics of porous carbon prepared at 800 °C for 2 h with various amounts of Fe. (**a**) Adsorption isotherms of MB, (**b**) adsorption kinetics of MB, (**c**) adsorption isotherms of MO, (**d**) adsorption kinetics of MO.

**Table 1 nanomaterials-09-01098-t001:** Porosity parameters of as-prepared porous carbon prepared with various amounts of Fe.

Sample	S_BET_ (m^2^/g)	V_total_ (cm^3^/g)	V_meso_ (cm^3^/g)	V_meso_/V_total_ (%)
0 Fe	753	0.453	0.141	31.1
0.5 Fe	817	0.483	0.173	35.8
1.0 Fe	1078	0.636	0.375	59.0
1.5 Fe	903	0.586	0.357	60.9
2.0 Fe	767	0.484	0.183	37.8

**Table 2 nanomaterials-09-01098-t002:** Kinetics parameters calculated on the basis of Equations (2) and (3) for methylene blue (MB) adsorption of porous carbon prepared with various amounts of Fe.

Sample	Dyes	Langmuir			Freundlich			Pseudo-First-Order			Pseudo-Second-Order		
		Q_0_	B	R^2^	K_F_	N	R^2^	Q_e_	K_1_	R^2^	Q_e_	K_2_	R^2^
**0 Fe**	MB	188.4	0.031	0.83	1.6	1.15	0.91	124.8	0.50	0.995	142.9	0.016	0.999
**0.5 Fe**		412	0.012	0.94	1.5	1.01	0.95	240	0.59	0.992	250	0.012	0.999
**1.0 Fe**		506.8	0.009	0.96	1.4	0.99	0.96	274.7	0.59	0.995	333.3	0.009	0.998
**1.5 Fe**		442.5	0.011	0.95	1.4	1.0	0.95	260.9	0.41	0.993	277.8	0.005	0.999
**2.0 Fe**		194	0.029	0.84	1.6	1.2	0.91	127	0.50	0.996	142.9	0.012	0.998

**Table 3 nanomaterials-09-01098-t003:** Kinetics parameters calculated on the basis of Equations (2) and (3) for methyl orange (MO) adsorption of porous carbon prepared with various amounts of Fe.

Sample	Dyes	Langmuir			Freundlich			Pseudo-First-Order			Pseudo-Second-Order		
		Q_0_	B	R^2^	K_F_	N	R^2^	Q_e_	K_1_	R^2^	Q_e_	K_2_	R^2^
**0 Fe**	MO	249.5	0.020	0.95	1.7	1.16	0.91	161.5	0.58	0.998	166.7	0.016	0.999
**0.5 Fe**		559	0.008	0.98	1.5	1.02	0.95	311.9	0.50	0.995	333.3	0.005	0.999
**1.0 Fe**		683.8	0.007	0.96	1.1	0.99	0.94	370.7	0.60	0.995	379	0.007	0.998
**1.5 Fe**		521	0.007	0.96	1.4	1.0	0.95	334.1	0.56	0.993	344.8	0.006	0.999
**2.0 Fe**		254	0.020	0.98	1.8	1.2	0.90	166.8	0.51	0.996	172.4	0.013	0.999

**Table 4 nanomaterials-09-01098-t004:** Comparison of dye adsorption capacity of hierarchical porous carbon prepared in this work and other adsorbents reported in the literature.

Adsorbent	Dyes	T (°C)	BET (m^2^/g)	Q_m_ (mg/g)	Reference
Mesoporous activated carbon	MB	30	1135	359	[44]
Activated carbon	MB	30~50	1940	434.78	[45]
Activated carbon	MB	30~50	1060	102.04	[3]
Activated carbon	MG	25~50	1000	149	[46]
Porous carbon	RB	-	2721	479	[47]
Hierarchical porous carbon materials	MB/DB	25	1913	1585.7/438.6	[48]
Hierarchical porous carbon materials	MO	25	1338.9	598.8	[49]
Hierarchical porous carbon materials	MB/MO	25	1078	506.8/683.8	This work

BET: Brunauere Emmette Teller, RB: Rhodamine B, DB: Direct black 38, MG: Malachite green.

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
