# Peer review of "Synthesis of Hierarchical Porous Carbon in Molten Salt and Its Application for Dye Adsorption"

_nanomaterials, 2019, doi:10.3390/nano9081098_

Round 1

Reviewer 1 Report

This paper describes the facile synthesis of porous carbon via molten salt technique.  Optimum amount of oxidizer (iron nitrate) was identified as giving highest SA, highest mesoporosity, and maximal adsorption of methylene blue and methyl orange.  Adsorption kinetics were found to follow pseudo second order kinetic model.

The data is properly presented, and conclusions are sound.  Comparison of results is made with previous results in the literature, although little discussion is devoted to explanation of superiority of authors’ technique.  Care should be taken on assuming the scalability of the process.  Reader might be interested in scalability of the process, as there can be large differences in product comparing small batches to large batches in MSS.

The manuscript is well-written, with only a few spelling errors. For example, on page 5, “hiring” vs “firing”, and “board” vs “broad”.  Fig. 5 should contain more info in figure caption, as small type below (c-e) is not legible. 

Author Response

Thank you very much for your consideration of our manuscript. We would also like to thank all the reviewers for their valuable comments and suggestions, based on which, we have revised our manuscript carefully. All the modified parts are highlighted in the revised manuscript. The detailed responses to your and reviewers’ comments are listed below. We hope that we have addressed the issues raised by you and the reviewers properly and the revised manuscript is now acceptable for publication.

The texts written in blue are our responses to the questions raised by the reviewers.

The texts written in red in the revised manuscript highlight the corrections/modifications according to reviewers’ comments.

Reviewer(s)' Comments to Author:
Reviewer 1: 

This paper describes the facile synthesis of porous carbon via molten salt technique. Optimum amount of oxidizer (iron nitrate) was identified as giving highest SA, highest mesoporosity, and maximal adsorption of methylene blue and methyl orange.  Adsorption kinetics were found to follow pseudo second order kinetic model.

The data is properly presented, and conclusions are sound. Comparison of results is made with previous results in the literature, although little discussion is devoted to explanation of superiority of authors’ technique. Care should be taken on assuming the scalability of the process. Reader might be interested in scalability of the process, as there can be large differences in product comparing small batches to large batches in MSS.

Response: Thank you for your comments. The molten salt method has a wide range of applications and can be used to prepare various materials. Besides, the molten salt method (MSS) has the advantage of easy isolation of the product and has in fact a long history as a solvent in research as well as in industry[R1-R3]. Therefore, there might be no differences in large batches as well as small batches.

The manuscript is well-written, with only a few spelling errors. For example, on page 5, “hiring” vs “firing”, and “board” vs “broad”.  Fig. 5 should contain more info in figure caption, as small type below (c-e) is not legible.

Response: Thank you. Following on this comment, done as suggested.

The revised captions have been included in the revised manuscript and are shown below.

Fig. 5 TEM and EDS results of the sample prepared with 1.0 wt% Fe. (a) low-resolution image; (b) high-resolution image; (c) mapping scan of C in red box; (d) mapping scan of Fe in red box; (e) mapping scan of O in red box

[R1] Liu X , Fechler N , Antonietti M . Salt melt synthesis of ceramics, semiconductors and carbon nanostructures[J]. Chemical Society Reviews, 2013, 42(21):8237.

[R2] Huo K , Zhu B , Fu J , et al. Large-Scale Synthesis of Mullite Nanowires by Molten Salt Method[J]. Journal of Nanoscience and Nanotechnology, 2010, 10(7):4792-4796.

[R3] Lin C , Chi Y , Jin Y , et al. Experimental Study on Molten Salt Oxidation of High Salt Content Pharmaceutical Residue ☆[J]. Procedia Environmental Sciences, 2016, 31:335-344.

Reviewer 2 Report

Authors carefully demonstrated the importance of molten salt methods and its applications

The following minor revisions are needed to publish above paper in nanomaterials

1. It is not so clear why authors used ball milling method initially to mix the reactants , based on authors experience Fe2O3 easier to prepare by simple mixing ( for example : MSM : Electrochimica Acta  118 (2014) 75), Materials Letters 212(2018) 186 probably authors used well mixing of carbon and also some XRD patterns of low intensity, probably XRD recorded due to fast scan   and discuss the formation mechanism  of Fe3O4 (The Journal of Physical Chemistry C 121 (7), 3778), Journal of Alloys and Compounds 565(2013)6054. 

Nice to do the XRD at slow scan, if possible include the lattice parameter values

2. Nice to discuss previous photo catalytic studies of the compounds (Journal of Alloys and Compounds 671,(2016) 552, Materials Science in Semiconductor Processing 40(2015) 194

3. Introduction section: Since this paper is on molten salt method authors mention Molten salt method important method : instead of discussing other method, nice to expand the discussion of molten salt method over preparation of other materials  (Electrochemical and Solid State Letters 14 (2011)A79, RSC Advances 2(2012)9619, Materials Letters 140 (2015)115, ACS Applied Materials and Interfaces" 5 (20) (2013) 9957, ACS Applied Materials and Interfaces 5(10)(2013)4361, Journal of Physical Chemistry C119(9) (2015) 4709, Electrochimica Acta  118 (2014) 75 , RSC Advances 5(37)(2015)29535), Journal of the Electrochemical Society 159(6) (2012)A762, Electrochimica Acta 182(2015)1060, Journal of Power Sources 225(2013)374, CrystEngComm 15(2013)3568, J. Power Sources 147(2005) 241, J. Power Sources 159 (2006) 263, J. Power Sources 160(2006) 1369, ACS Sustainable Chemistry & Engineering (2015)10.1021/acssuschemeng.5b00439, Electrochimica Acta 128(2014)192, Materials Letters 138 (2015)231, Electrochimica Acta 71(2012)227 to be included and discussed

4  growth mechanism and morphology should be expanded, morphology  are varies with molten salt, preparation, temp. , initial reactants etc Please read above references (comment : 3) and accordingly summarize

Author Response

 Thank you very much for your consideration of our manuscript. We would also like to thank all the reviewers for their valuable comments and suggestions, based on which, we have revised our manuscript carefully. All the modified parts are highlighted in the revised manuscript. The detailed responses to your and reviewers’ comments are listed below. We hope that we have addressed the issues raised by you and the reviewers properly and the revised manuscript is now acceptable for publication.

The texts written in blue are our responses to the questions raised by the reviewers.

The texts written in red in the revised manuscript highlight the corrections/modifications according to reviewers’ comments.

Reviewer(s)' Comments to Author:

1. It is not so clear why authors used ball milling method initially to mix the reactants, based on authors experience Fe2O3 easier to prepare by simple mixing ( for example : MSM : Electrochimica Acta  118 (2014) 75), Materials Letters 212(2018) 186 probably authors used well mixing of carbon and also some XRD patterns of low intensity, probably XRD recorded due to fast scan and discuss the formation mechanism of Fe3O4 (The Journal of Physical Chemistry C 121 (7), 3778), Journal of Alloys and Compounds 565(2013)6054.

Nice to do the XRD at slow scan, if possible include the lattice parameter values

Response: Thank you for your comments. In the experiment, the starting materials were just milled together in mortar and not used ball milling method. Fig. R1 was XRD pattern of as-prepared samples with various amounts of Fe(NO3)3·9H2O with slow scan rate. Compared with Figure 1, the peak strength of Fe2O3 and Fe3O4 has little change. According to the XRD results, the obtained lattice parameter values for Fe2O3 were respectively a=13.21 Å, b=9.78 Å and c=8.37 Å, and a=6.21 Å, b=5.89 Å and c=14.77 Å for Fe3O4. In the XRD pattern (Fig.1), the peak intensity of Fe2O3 and Fe3O4 were low, which might also attribute to that the amounts of Fe(NO3)3·9H2O added was limited. In the preparation process, because of the reaction was carried out in Ar atmosphere, the Fe(NO3)3·9H2O decomposed into Fe2O3 firstly and then some Fe2O3 became into Fe3O4 by reductive carbon coming from glucose in firing[R1-R2].

Fig. R1 XRD pattern of as-prepared samples with various amounts of Fe(NO3)3·9H2O. (ICDD: 00-016-0653 (Fe2O3) and 01-076-0956(Fe3O4)).

The revised captions have been included in the revised manuscript in line 13~14, page 5 and are shown below.

Apart from the amorphous carbon, Fe2O3 and Fe3O4 were identified in the samples using Fe(NO3)3·9H2O, suggesting the decomposition of Fe(NO3)3·9H2O into Fe2O3 firstly and then some Fe2O3 became into Fe3O4 by reductive carbon come from glucose on firing[32-33]. The obtained lattice parameter values for Fe2O3 were respectively a=13.21 Å, b=9.78 Å and c=8.37 Å, and a=6.21 Å, b=5.89 Å and c=14.77 Å for Fe3O4.

2. Nice to discuss previous photo catalytic studies of the compounds (Journal of Alloys and Compounds 671, (2016) 552, Materials Science in Semiconductor Processing 40(2015) 194

Response: Thank you for your comments. According to your suggestions, Fe2O3 or NiO nanoparticles could be used in energy storage, electrochemical sensing or photo catalytic et al. In spite of Fe2O3 existed in the sample, the amounts of them was limited. If they are to be used in these areas, their content needs to be increased. Thank you for your suggestion, we will study it in the future.

3. Introduction section: Since this paper is on molten salt method authors mention Molten salt method important method : instead of discussing other method, nice to expand the discussion of molten salt method over preparation of other materials  (Electrochemical and Solid State Letters 14 (2011)A79, RSC Advances 2(2012)9619, Materials Letters 140 (2015)115, ACS Applied Materials and Interfaces" 5 (20) (2013) 9957, ACS Applied Materials and Interfaces 5(10)(2013)4361, Journal of Physical Chemistry C119(9) (2015) 4709, Electrochimica Acta  118 (2014) 75 , RSC Advances 5(37)(2015)29535), Journal of the Electrochemical Society 159(6) (2012)A762, Electrochimica Acta 182(2015)1060, Journal of Power Sources 225(2013)374, CrystEngComm 15(2013)3568, J. Power Sources 147(2005) 241, J. Power Sources 159 (2006) 263, J. Power Sources 160(2006) 1369, ACS Sustainable Chemistry & Engineering (2015)10.1021/acssuschemeng.5b00439, Electrochimica Acta 128(2014)192, Materials Letters 138 (2015)231, Electrochimica Acta 71(2012)227 to be included and discussed

Response: Thank you for your comments. The molten salt method has a wide range of applications and can be used to prepare various materials. We will supplement it in the introduction section.

The revised captions have been included in the revised manuscript in line 22~23, page 2 and are shown below.

Recently, molten salt synthesis method applicable to preparation of a wide range of nanomaterials including binary, ternary/multinary oxides[16-25], hydroxide[26], non-oxides[27] and porous carbon materials[28-31].

4 growth mechanism and morphology should be expanded, morphology are varies with molten salt, preparation, temp. , nitial reactants etc. Please read above references (comment : 3) and accordingly summarize

Response: Thank you for your comments. Generally speaking, the formation mechanism of molten salt method for preparing materials are very complex as they depend on temperature and nature of the molten salt and metal ions. Research results of many published literatures indicated that the as-prepared samples were mainly nano-materials and had higher specific surface areas. However, the morphology and average sizes sometimes vary with different reaction temperatures and molten salt types. The specific reaction mechanism is not very clear. The molten salt method makes the reaction in solution environment, in general, the physics and chemistry of salt melts, the operating temperature and the solubility of the starting materials will influence the morphology and size of the samples. For different samples, the growth mechanism is also different.

[R1] Petnikota S, Maseed H, Srikanth V, et al. Experimental elucidation of a graphenothermal reduction mechanism of Fe2O3: an enhanced anodic behavior of an exfoliated reduced graphene oxide/Fe3O4 composite in Li-ion batteries[J]. The Journal of Physical Chemistry C, 2017, 121(7): 3778-3789.

[R2] Das B, Reddy M V, Chowdari B V R. Li-storage of Fe3O4/C composite prepared by one-step carbothermal reduction method[J]. Journal of Alloys and Compounds, 2013, 565:90-96.